# Effectiveness of a third BNT162b2 mRNA COVID-19 vaccination during pregnancy: a national observational study in Israel

Joshua Guedalia [1,6], Michal Lipschuetz [1,2,6], Ronit Calderon-Margalit[3], Sarah M. Cohen [1], Debra Goldman-Wohl[1], Tali Kaminer[4], Eli Melul[4], Galit Shefer[4], Yishai Sompolinsky[1], Asnat Walfisch [5], Simcha Yagel[1] & Ofer Beharier [1] ✉

The Centers for Disease Control (CDC) recommend a third dose of COVID-19 vaccine for pregnant women, although data regarding effectiveness during pregnancy are lacking. This national, population-based, historical cohort study of pregnant women in Israel, delivering between August 1, 2021 and March 22, 2022, aims to analyze and compare the third and second doses' vaccine effectiveness in preventing COVID-19-related hospitalizations during pregnancy during two COVID-19 waves (Delta variant in the summer of 2021 and Omicron, BA.1, variant in the winter of 2022). Time-dependent Cox proportional-hazards regression models estimate the hazard ratios (HR) and 95% confidence intervals (CI) for COVID-related outcomes according to vaccine dose, and vaccine effectiveness as 1-HR. Study includes 82,659 and 33,303 pregnant women from the Delta and Omicron waves, respectively. Compared with the second dose, the third dose effectively prevents overall hospitalizations with SARS-CoV-2 infections, with estimated effectiveness of 92% (95% CI 83–96%) during Delta, and enhances protection against significant disease during Omicron, with effectiveness of 92% (95% CI 26–99%), and 48% (95% CI 37–57%) effectiveness against hospitalization overall. A third dose of the BNT162b2 mRNA COVID-19 vaccine during pregnancy, given at least 5 months after the second vaccine dose, enhances protection against adverse COVID-19-related outcomes.

Millions of pregnant women have been infected with SARS-CoV-2 since the World Health Organization declared the COVID-19 a global pandemic more than two years ago. Pregnancy has been shown to considerably increase the risk for severe COVID-19 illness, mechanical ventilation, and death, as compared to age-matched non-pregnant women. Moreover, SARS-CoV-2 infection during pregnancy has also

been associated with poor obstetric outcomes, including preterm birth and stillbirth[1–4].

Despite the threat posed by COVID-19, pregnant women were excluded from the initial COVID-19 vaccine trials, leading to substantial knowledge gaps on the effect of vaccines on maternal and fetal health. Nevertheless, the urgent need to protect this vulnerable population

[1]Obstetrics & Gynecology Division, Hadassah Medical Center, Faculty of Medicine of the Hebrew University of Jerusalem, Jerusalem, Israel. [2]Henrietta Szold Hadassah Hebrew University School of Nursing in the Faculty of Medicine, Jerusalem, Israel. [3]Braun School of Public Health, Hadassah Medical Center, Faculty of Medicine of the Hebrew University of Jerusalem, Jerusalem, Israel. [4]TIMNA-Israel Ministry of Health's Big Data Platform, Israel Ministry of Health, Jerusalem, Israel. [5]Helen Schneider Hospital for Women, Rabin Medical Center, Petah Tikva, and Sackler Faculty of Medicine, Tel Aviv University, Tel Aviv, Israel. [6]These authors contributed equally: Joshua Guedalia, Michal Lipschuetz. ✉e-mail: oferbeharier@gmail.com

dictated the inclusion of pregnant women in vaccination campaigns, before clinical trials were completed[2,5–9]. Real-world data confirming the safety and effectiveness of COVID-19 vaccines during pregnancy are critical to support public health policy. Indeed, essential data have accumulated regarding the effectiveness and safety of the BioNTech BNT162b2 mRNA COVID-19 vaccine in pregnant women[6–9]. Two vaccine doses in pregnancy appeared to protect against infection without increase in prenatal or early neonatal morbidity[4,6,7], however early studies did not establish vaccine effectiveness against significant disease. As vaccine-induced immune protection appeared to wane, health organizations proposed booster vaccination with a third vaccine dose, including during pregnancy, without an evidence basis for the necessity and effectiveness of a third dose in this population[10,11]. Indeed, following a surge in COVID cases in the summer of 2021, Israel launched a population-wide booster vaccination campaign using BNT162b2 mRNA vaccines, calling for all persons over the age of 16 years who had received their second dose at least 5 months prior, to present for a third dose. This campaign included pregnant women starting in August 2021.

We conducted a nationwide historical cohort study to evaluate the evidence regarding this vaccine strategy. We investigated the effectiveness of the three-dose vaccine regimen in mitigating significant disease in pregnant women during two periods of the COVID-19 pandemic: the summer outbreak, during which the Delta variant was dominant (1 August 2021 to 1 December 2021), and the winter outbreak, during which the Omicron BA.1 variant became dominant (15 December 2021 to 22 March 2022; Fig. 1)[12]. These variants (Delta and Omicron) substantially differed from one another in virulence[13] and their ability to evade vaccine-mediated immune protection[14].

## Results

During the study period from 1 August 2021 to 22 March 2022, a total of 82,803 pregnant women delivered and met the eligibility criteria (Methods section; see flow chart Fig. 2). Characteristics of the study population at delivery are presented in Table 1. The third dose group was older (23.8% between 36 and 45 years old) than the second dose or unvaccinated groups (18.0% and 15.6%, respectively). The proportion of primiparous women among the third dose group was higher than

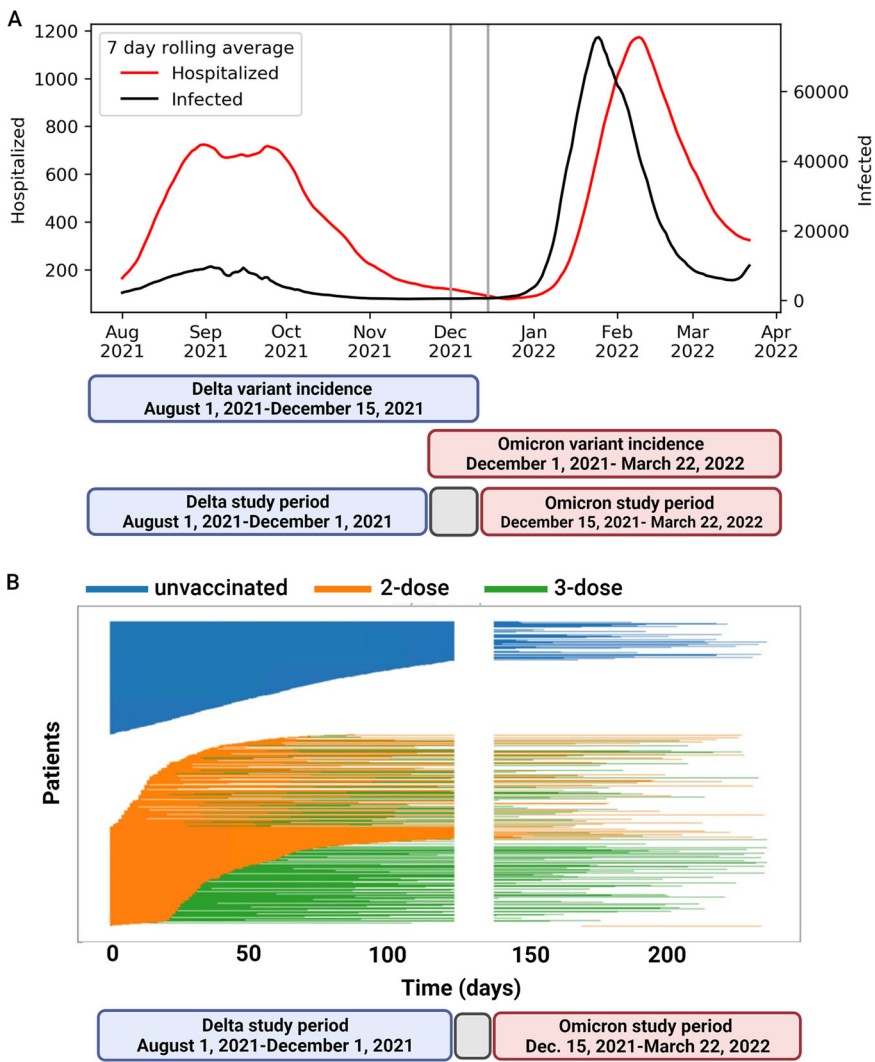

**Fig. 1 | Israel national data of confirmed SARS-CoV-2 infections, and cases of severe COVID-19 during the study timeline, and graphic representation of participants follow-up. A** Weekly incidence numbers, of all Israeli population, SARS-CoV-2 infection (Black line) and severe COVID-19 (Red line), on different scales, between August 2021 and April 2022; the lines are the point estimate of the weekly incidence events. The study periods are demonstrated beneath the graph (Delta period in blue and Omicron in red; the gray represents the time frame excluded from the study due to overlap of the variants). **B** Sample of participant follow-up time during the two study periods. Each row represents a participant, colored by their vaccination status during follow-up. Third vaccine (green line), second vaccine (orange line), and unvaccinated (blue line). Participants change color when they move between groups.

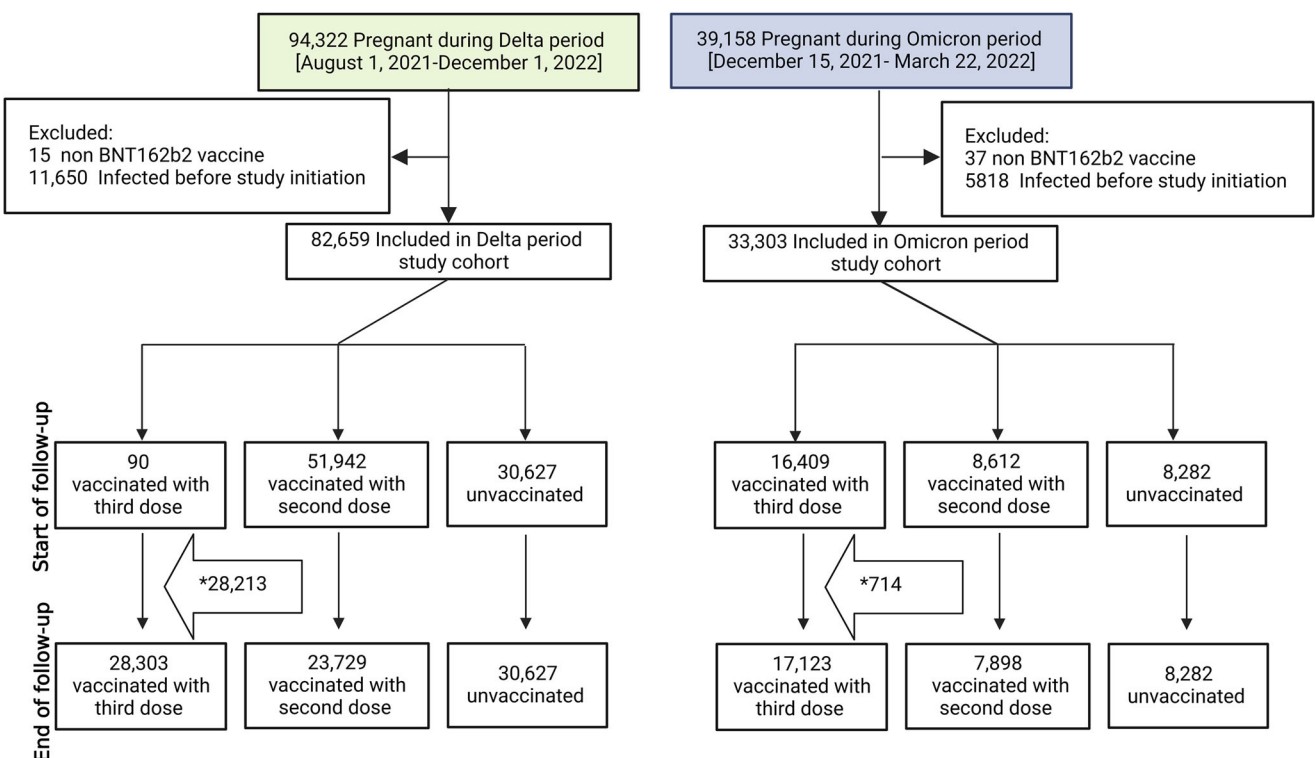

**Fig. 2 | Study population and flow chart of cohort selection.** The flow chart shows the three comparison groups investigated during the two study periods. Horizontal arrows show the number of participants that received their booster vaccinations during the study period.

that of the second dose or unvaccinated groups (29.5% vs. 26.9% and 26.8%, respectively). Higher rates of grand-multiparity were observed among the unvaccinated group than the third and second dose groups (21.4% vs 10.2% and 12.9%, unvaccinated, vs third and second dose,

respectively). The unvaccinated group included 10,749 (35.1%) who had no documentation for any SARS-CoV-2 testing, compared to 3,484 (15.2%) and 2,964 (10.1%) of pregnant women in the second and third dose groups, respectively.

During the Delta period, 10 (0.04%) hospitalizations with COVID-19 were documented in the third dose group, 105 (0.20%) in the second dose group, and 341 (1.11%) in the unvaccinated group. During the Omicron period, 260 (1.5%), 217 (2.5%), and 207 (2.5%) were hospitalized, respectively; reflecting the greater transmissibility of the Omicron variant (Suppl. Table 1). Cumulative risk curves for hospitalizations with a diagnosis of SARS-CoV-2 infection are shown in Fig. 3. The figure shows that over time, the rate of hospitalization of women in the third dose group was consistently lowest of the three groups, and that the second dose was effective in preventing hospitalization during the Delta period but not during the Omicron period. Suppl Fig. 1 shows cumulative risk curves for hospitalization with significant disease, and with severe disease.

Table 2 presents the HR and estimated vaccine effectiveness (1-HR%) for the various study outcomes, by vaccine dose. Compared with unvaccinated women, the third dose vaccine effectiveness was estimated to be 97% (95%, CI 95–99%) and 43% (95%, CI 31–53%) for hospitalization with SARS-CoV-2 infection during the Delta and Omicron periods, respectively. The effectiveness of the third dose in preventing significant disease was 99% (95%, CI 93–100%) and 97% (95%, CI 72–100%), during the Delta and Omicron periods, respectively. Similar estimates were evident in preventing severe disease (Table 2 and 99% (95%, CI 89–100%) and 94% (95%, CI 43–99%) for severe disease, compared to the unvaccinated group, during the Delta and Omicron periods, respectively). Being vaccinated with a second dose provided high protection ≥5 months following vaccination during the Delta period, with an effectiveness of 97% (95% CI 92–99%) against

**Table 1 | Characteristics of the study cohort by vaccination status at the time of delivery**

| Study period [1 August 2021–22 March 2022] (N = 82,809) | | | |
|---|---|---|---|
| | **3-Dose** | **2-dose** | **Unvaccinated** |
| *n* % | 29,331 (35.4) | 22,862 (27.6) | 30,616 (37) |
| Maternal age in years | | | |
| <18–26 | 4754 (16.3) | 5991 (26.3) | 11,897 (39.0) |
| 27–35 | 17,489 (59.9) | 12,710 (55.7) | 13,864 (45.4) |
| 36–45 | 6984 (23.8) | 4103 (18.0) | 4769 (15.6) |
| Multifetal delivery | 590 (2.0) | 402 (1.8) | 552 (1.8) |
| Parity | | | |
| Primipara | 8642 (29.5) | 6149 (26.9) | 8200 (26.8) |
| Multipara (2–4) | 17,710 (60.4) | 13,773 (60.2) | 15,853 (51.8) |
| Grandmultipara (5+) | 2979 (10.2) | 2940 (12.9) | 6563 (21.4) |
| Number of SARS-CoV-2 PCR/antigen tests | 3.9 (±1.7) | 3.6 (±1.9) | 2.5 (±2.1) |
| 0 | 2964 (10.1) | 3484 (15.2) | 10749 (35.1) |
| 1 | 137 (0.5) | 127 (0.6) | 298 (1.0) |
| 2 | 3439 (11.7) | 3459 (15.1) | 6314 (20.6) |
| 3 | 198 (0.7) | 149 (0.7) | 164 (0.5) |
| 4 | 3832 (13.1) | 3241 (14.2) | 3663 (12.0) |
| ≥5 | 18761 (64.0) | 12402 (54.2) | 9428 (30.8) |

Data are *n* (%), and mean (±standard deviation); data are calculated according to the vaccine status of women at delivery.

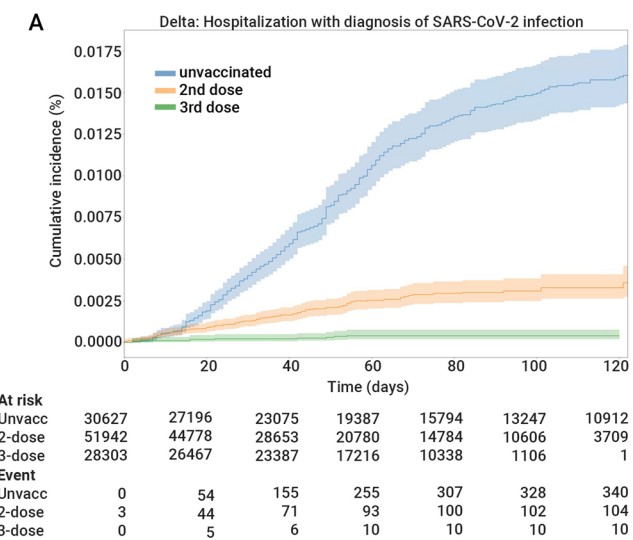

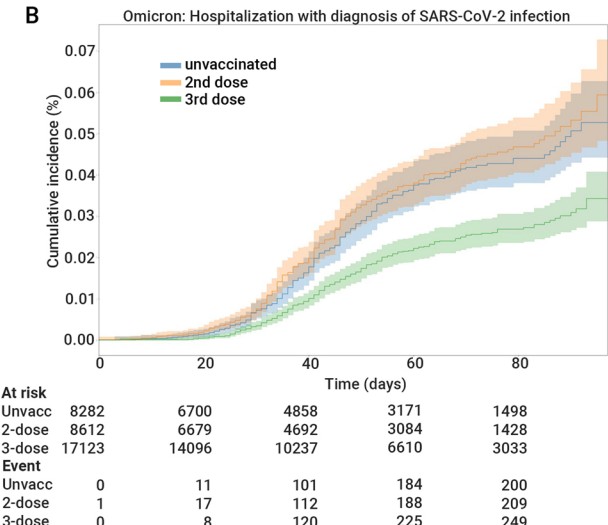

**Fig. 3 | Cumulative incidence of study outcomes, according to COVID-19 waves and vaccination status.** Cumulative incidence curves comparing the two study periods (Delta period on the left (**A**) and Omicron period on the right (**B**)) for hospitalizations with a diagnosis of SARS-CoV-2 infection in pregnant women, by vaccination status (Third vaccine-green line, second vaccine-orange line, and unvaccinated- blue line). The main line is the point estimate of the cumulative incidence and the shaded areas represent 95% confidence intervals. The number at risk at each time point and the cumulative number of events are also shown for each outcome.

significant disease, and 96% (95% CI 86–99%) against severe disease. During the Omicron period, the second vaccine was not effective in protecting against all study outcomes. The estimated effectiveness of the third vs. the second dose (additional protection) against hospitalization with SARS-CoV-2 infection was 92% (95% CI, 83–96%) during the Delta period, and 48% (95% CI, 37–57%) during the Omicron period. During the Delta period the contribution of the third dose, in addition to the second dose, for protection against significant disease was modest and could not be calculated, as the numbers of affected cases were small. In contrast, the third dose provided considerably enhanced protection during the Omicron period with effectiveness of 92% (95% CI 26–99%) in addition to the second dose.

In order to access whether the length between the 2nd and 3rd dose affected the vaccine protection we further analyzed the data. When calculating the risk for COVID-19 hospitalization during the Omicron period by intervals of 30 days and we found that the time elapsed between vaccination had minimal to no effect on the risk for hospitalization (there was insufficient incidence during the Delta period for meaningful analysis). During the Omicron period, when stratified by intervals between vaccinations within the 30 days following eligibility, the risk for hospitalization was estimated as 1.36%, in the following 30 days – 1.46%, 1.55%, and 1.78%, for every sequential interval of 30 days, respectively.

## Discussion

Our study showed that a third dose of the BNT162b2 mRNA COVID-19 vaccine, given at least 5 months after the second vaccine dose, provides additional protection during pregnancy against hospitalizations with a diagnosis of SARS-CoV-2 infection, and against significant and severe disease. As described above, significant disease was defined by documented hospitalization with moderate COVID-19-related disease, or worse, i.e. COVID-19 related pneumonia justifying hospitalization. Severe disease was defined as a resting respiratory rate >30 breaths per minute, oxygen saturation on room air <94%, or as the need for mechanical ventilation and clinical severe organ failure.

The mRNA vaccines currently available were designed to prevent infection and disease from the wild type SARS-CoV-2 strains. Data from non-pregnant populations demonstrate that the effectiveness of the second vaccine dose declines over time as the humoral immunity wanes and new variants emerge[15–18]. Our data concur with these reports. Previous studies reported 98% effectiveness of the second dose against hospitalization, shortly after vaccination[3,19]. We detected reduced effectiveness more than 5 months after the second vaccine dose (61% during the Delta period and none in the Omicron period), findings that might support waning of immunity. In this context, the third dose provided additional protection during the Delta and Omicron periods (97% and 43% protection, respectively) when compared to unvaccinated patients, emphasizing the benefit of vaccine boosting.

We previously showed that a third dose of BNT162b2 mRNA vaccine significantly increased anti-SARS-CoV-2 antibody titers in maternal and cord blood[20]. In addition, a recent study found that a third booster dose was essential in building neutralizing antibody capacity against the Omicron variant among mothers and neonates[21]. These boosted antibody titers may have provided additional protection from the Delta variant and allowed protection from the Omicron variant.

When focusing on substantial COVID-19 illness, 5 months after the second dose, the second dose effectively protected against hospitalization complicated by significant disease (97%) and severe disease (96%) during the Delta period, but not during the Omicron period. The impact of the third boosting dose was substantial during the Omicron period, effectively protecting against hospitalization complicated by significant disease (97%) and severe disease (94%). To the best of our knowledge, our results present data regarding considerable vaccine effectiveness against severe COVID-19 disease during pregnancy. The fact that vaccines during pregnancy nearly abolish the risk for significant disease has been shown to play a role in patient decision-making regarding vaccination[22]. Hence, our study might contribute to promoting vaccination uptake among pregnant women.

In the present study, we focused on the impact of COVID-19 vaccine strategy on hospitalization with a diagnosis of SARS-CoV-2 infection, rather than population infection rates. Recorded infection rates may be biased by differential rates of testing in various population subgroups, most notably among unvaccinated patients. Indeed, our data show that unvaccinated pregnant women were considerably less likely to be tested (Table 1). However, while not uniformly executed in all maternity units, routine SARS-CoV-2 testing during maternity admissions was mandatory in most hospitals in Israel. Given the unbiased approach to testing, a finding of positive SARS-CoV-2

**Table 2 | Vaccine effectiveness measures**

| | Delta period | | | | | | Omicron period | | | | | |
|---|---|---|---|---|---|---|---|---|---|---|---|---|
| | Hospitalization | | Significant disease | | Severe disease | | Hospitalization | | Significant disease | | Severe disease | |
| | HR | 1-HR (95% CI) | HR | 1-HR (95% CI) | HR | 1-HR (95% CI) | HR | 1-HR (95% CI) | HR | 1-HR (95% CI) | HR | 1-HR (95% CI) |
| 3rd vs. 2nd vaccine group | 0.08 (0.04–0.17) | 92% (83–96) | 0.15[a,b] (0.01–1.83) | 0 vs. 4 | 0.17[a,b] (0.01–2.40) | 0 vs. 3 | 0.52 (0.43–0.63) | 48% (37–57) | 0.08 (0.01–0.74) | 92% (26–99) | 0.41[a,b] (0.02–6.68) | 0 vs. 1 |
| 2nd dose vs. unvaccinated group | 0.39 (0.31–0.49) | 61% (51–69) | 0.03 (0.01–0.08) | 97% (92–99) | 0.04 (0.01–0.14) | 96% (86–99) | 1.12 (0.92–1.36) | [-12%] ([-36]–8) | 0.49 (0.16–1.47) | 51% ([-47]–84) | 0.17 (0.02–1.47) | 83% ([-47]–98) |
| 3rd dose vs. unvaccinated group | 0.03 (0.01–0.05) | 97% (95–99) | 0.01[a] (0.00–0.07) | 99% (93–100); 0 vs. 108 | 0.02[a] (0.00–0.11) | 99% (89–100); 0 vs. 64 | 0.57 (0.47–0.69) | 43% (31–53) | 0.03 (0.00–0.28) | 97% (72–100) | 0.06[a] (0.01–0.57) | 94% (43–99); 0 vs. 5 |

HR of COVID-19 hospitalization degree for vaccinated group vs. unvaccinated group and between the vaccinated groups, in two study periods [Delta period- 1 August 2021–1 December 2021; Omicron period- 15 December 2021–22 March 2022]. The study period populations were 82,659 and 33,303 in each of the two periods, respectively. In each study period, 28,213 and 714 individuals were first included in the 2-dose group and then re-recruited to the 3-dose group.

HR calculated using Cox proportional-hazards regression model with time-dependent covariates controlling for maternal age, parity and days of follow-up.

Outcomes definitions: hospitalization with SARS-CoV-2 infection; significant disease (e.g., COVID-19-related pneumonia associated with COVID-19 justifying hospitalization), and severe disease (e.g., resting respiratory rate >30 breaths per minute, O2 saturation on room air <94%, etc.).

aA single case was imputed to allow estimations of HR. Estimation is based on the mean of 1000 simulations.

bEstimates (1-HR) were only calculated for cells with 5 events or more, otherwise, raw counts are reported.

during hospitalization represents a better sensor for infection burden, and we therefore assessed and analyzed the data accordingly.

Most previous reports analyzed pregnancy data from a single COVID-19 wave, narrowing observations. We analyzed data from two discrete periods, when two variants having different characteristics were dominant. We focused the time margins on the periods dominated by the Delta and Omicron variants, to present a more comprehensive understanding of vaccine and boosting effectiveness on different viral variants. Indeed, we found substantial differences between the two time periods, which might reflect differences in virulence, ability to evade vaccine-mediated immune protection, and waning of protective titers over time.

The CDC and other health organizations now recommend COVID-19 vaccination for pregnant women to reduce the risks of severe disease and complications[11,23]. Similar to the general population, these recommendations include boosting of pregnant women with a third vaccine dose, 5 months following the second vaccine dose. Our findings provide insight into the impact of COVID-19 vaccines during pregnancy and the advantage conferred by the third, boosting dose against serious illness, and serve to reinforce recommendations to vaccinate and boost this population, providing clinicians and policymakers with essential evidence to inform consultation and decision-making.

Research into the long-term consequences of COVID-19 in pregnancy for mothers and newborns is still scarce. Initial, preliminary reports of the long-term effects of maternal COVID-19 infection during pregnancy have suggested worrying adverse neurodevelopmental sequelae[24–26]. These cases highlight the urgent need for data on measures to limit maternal infection, which may have as yet unknown adverse consequences.

Our study has several limitations. Since the study is based on real-world collected data, no randomization was possible. Individuals opting to decline boosting doses or to refuse vaccination or evade testing may differ in risk-taking behavior, demographic or obstetric characteristics from those opting in. Moreover, vaccinated groups may behave in a more cautious manner that could reduce the chances of infection regardless of vaccination. In addition, natural and hybrid immunity acquired in patients over the course of the pandemic, which may not have been captured by testing in the community, may have had differential protective effects in unvaccinated and vaccinated groups. These are possible sources of bias that are difficult to adjust for in a study like ours, but must be stated and accounted for. We also recognize that other, unaccounted-for individual and group differences in risk factors for severe illness or the likelihood of exposure to the virus, may have impacted our results. Our findings were limited to the BNT162b2 vaccine. We cannot infer whether our observations are relevant to preventing reinfection in convalescent COVID-19 pregnant women, or populations administering other vaccines. The decision to vaccinate during pregnancy is a balance between benefit and effectiveness vs. safety. We did not evaluate COVID-19 vaccine safety, however several other large studies have done so[3,27], and demonstrated a favorable safety profile. Finally, we lack data on variant sequencing of the diagnostic tests used in this study. However, the inclusion of two time periods dominated by distinct variants strengthens our findings.

When compared to eligible non-boosted or unvaccinated pregnant women, those who received a third dose of BNT162b2 had a lower incidence of hospitalization with SARS-CoV-2 infection during the Delta period and considerably lower incidence of COVID-19 related outcomes during the Omicron period. Our data and analyses provide the necessary evidence to support current recommendations to vaccinate pregnant women with the third boosting dose of COVID-19 vaccine.

## Methods
### Study design
In Israel, the Ministry of Health (MOH) has collected information on all SARS-CoV-2 PCR and all institutionally conducted antigen tests since

the beginning of the pandemic. The MOH database maintains information on hospitalizations, severity of cases, and outcomes of patients with confirmed COVID-19. MOH also routinely collects information on all births in Israel. The current study is based on linkage of these datasets.

The study cohort included women who had a documented delivery between August 1, 2021 (the date on which a third boosting dose of Pfizer BNT162b2 mRNA vaccines became available for the younger population, including pregnant women) to 22 March 2022.

The study included unvaccinated pregnant women and those eligible to receive a third dose (≥150 days from the date they received their second dose) during the study periods. Women who had a documented positive SARS-CoV-2 test prior to follow-up time or had received one vaccine or a fourth boosting dose, were not included in this study.

The study protocol was approved by the Helsinki Committee of the Hadassah Medical Center. The committee granted exemption from informed consent, based on preserving the participants' anonymity.

## Study population
Three groups were compared for each time period: Group 1 included eligible pregnant women who received, prior to or during the given study period, a third boosting dose (third dose group); Group 2 were pregnant women who were eligible prior to or during the study period for a third boosting dose, but did not receive it (second dose group); and Group 3 included women who were unvaccinated (unvaccinated group).

## Study covariates
For each participant in the study, the following sociodemographic data were extracted: maternal age, parity (primipara- first delivery; multipara- from second to fourth delivery and grandmultipara- fifth delivery or greater), number of fetuses in the index pregnancy, and gestational age at delivery. The following clinical data were extracted: delivery date, vaccination dates, RT-qPCR or institutionally administered rapid antigen test dates and results, dates of hospital admission, discharge, disease severity, or death. During the study periods, institutional tests were free and readily available via self-referral. Widely advertised calls for population testing for indications including suspected community or household exposure as well as occupational directives, were issued.

## Study outcomes
Study outcomes included cases hospitalized for any indication, with incidental diagnosis of SARS-CoV-2 infection; hospitalization with significant illness from COVID-19; or hospitalization with severe COVID-19 disease, during pregnancy for each time period.

Any documented hospitalization in COVID-19 wards with a positive SARS-CoV-2 result, was considered as hospitalization with a diagnosis of SARS-CoV-2 infection. Significant disease was defined by documented hospitalization with moderate COVID-19-related disease as defined by the MOH, or worse, from the first day of hospitalization. Severe disease was defined by documented hospitalization with severe COVID-19-related disease (MOH) or worse, i.e., critical disease or death during the study period. MOH criteria defined moderate disease as COVID-19 related pneumonia justifying hospitalization; severe disease as a resting respiratory rate >30 breaths per minute, oxygen saturation on room air <94%, or ratio of PaO2 to FiO2 < 300; and critical disease as the need for mechanical ventilation and clinical severe organ failure[28].

## Follow-up time
The study was divided into two follow-up periods: The Delta period (1 August 2021 to 1 December 2021), and the Omicron period (15 December 2021 to 22 March 2022).

Eligible women were followed from the beginning of the study periods. Women who moved from one group to another (e.g. received a third dose) contributed follow-up time according to the time they were included in each group. Women were followed until delivery or incidence of a study outcome, whichever occurred first.

A total of 82,659 pregnant women contributed to the Delta period analysis and 33,303 to the Omicron period. Individual patients could be counted in both study periods in the course of their pregnancies ($n = 33,159$, see Fig. 1b).

## Statistical analysis
Descriptive statistics of the study population by vaccine dose at the end of follow-up is presented. Given that the independent variable (vaccine status) varied over time, univariate and multivariate survival analyses were performed with time-dependent covariates, in accordance with the study design, for each period separately. Kaplan–Meier analysis with a log-rank test was performed for univariate analysis. For each study period, time-dependent Cox proportional hazards models were constructed to estimate the hazard ratios (HR) and 95% confidence intervals (CI) for the study outcomes in the third dose group compared to the second dose group, controlling for maternal age and parity (model 1); additional models compared the third and second dose groups to the unvaccinated group, controlling for maternal age and parity (model 2). After estimation of HR for study outcomes in each group and study period, we calculated vaccine effectiveness as a percentage, defined as $100 \times (1 - HR)$; 95% confidence intervals were calculated similarly. To allow conservative estimations of HR and effectiveness in instances where there were no cases of study outcome in a study group, we imputed a single case, if the total number of cases was equal to or greater than 5. We simulated these imputations 1000 times to achieve robust estimates of the HR. This was used to estimate HR of the third dose group, for significant and severe disease during the Omicron period, and severe disease during the Delta period.

Python version 3.7.3 and lifelines 0.24.14 were used for multivariate survival analyses with time-dependent covariates. IBM-SPSS for Windows, version 24 (IBM Corp., Armonk, N.Y., USA), was used for the univariate analysis and all other statistical analyses. A $p$-value ≤ 0.05 was considered to indicate statistical significance in all analyses.

## Reporting summary
Further information on research design is available in the Nature Portfolio Reporting Summary linked to this article.

## Data availability
The raw national level data are protected and are not available due to data privacy laws. The data that support the findings of this study were provided by the Israel Ministry of Health but restrictions apply to the availability of these data, which were used under license for the current study, and so are not publicly available. Data are however available from the authors upon reasonable request and with permission of the Israel Ministry of Health.

## Code availability
The modeling in this paper used Python version 3.7.3 and lifelines 0.24.1, which are freely available.

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

## Acknowledgements
This work was supported by the "Ofek" Program of the Hadassah Medical Center. Figure 2 was created with BioRender.com.

## Author contributions
J.G., M.L., T.K., E.M., and O.B. saw the original data, collected it, and analyzed it. J.G., M.L., O.B., R.C.M., A.W., G.S., and S.Y. conceived and designed the study. J.G., M.L., O.B., R.C.M., S.M.C., D.W., and S.Y. wrote the manuscript. All authors critically reviewed the manuscript and decided to proceed with publication. R.C.M., S.Y., and O.B. supervised the study process. O.B. vouches for the data and analysis. T.K., E.M., and G.S. combined, anonymized, and QC of the MOH data.

## Competing interests
The authors declare no competing interests.
