## [Peer Review File · Nature Communications]

REVIEWER COMMENTS

Reviewer #1 (Remarks to the Author):

In this study, Dr. Guedalia and colleagues examined the potential benefit of a third dose of BNT162b2, an mRNA COVID-19 vaccine, in pregnancy. They note that such a third dose (or booster) is recommended for pregnant women, though there is not an established evidence base to support this recommendation, apart from extrapolating effectiveness findings from non-pregnant individuals.

As a randomized controlled trial cannot be done, the authors approach the problem by taking advantage of a large dataset available in Israel where vaccination history, pregnancy status, SARS-CoV-2 test results, and hospital outcomes are all linked. They also focused their analysis on two distinct periods defined by the predominant SARS-CoV-2 variant in circulation (e.g., Delta vs. Omicron BA.1) given the marked differences in the immune evasiveness of these variants.

The critical finding in this report is that a primary series (2 doses) was ineffective at preventing hospitalization with Omicron infection, and had much reduced effectiveness against hospitalization with moderate/severe illness compared to during the Delta wave. However, boosting with a 3rd mRNA dose rescued the protective effectiveness of vaccination, increasing protection against hospitalization with moderate/severe Omicron to 92%.

Strengths:

This is a critical study given (1) lack of data on boosting in pregnancy and (2) poor uptake of boosting across many populations. There is a dearth of understanding of how booster shots add significantly to a primary series for the prevention of bad outcomes in the setting of Omicron infection. While a primary series still provides a benefit on its own (particularly against severe disease and death), a primary series does not protect at all against Omicron infection (including mild and asymptomatic), for which the consequences in pregnancy may not be as benign. Robust data as presented in this report will be very helpful to underpin guidelines.

Weaknesses:

Like any effectiveness study, there is always the issue of the comparison groups not being quite the same people. Pregnant women who forgo the booster recommendation (and certainly those who don't get vaccinated at all) are different from those that follow recommendations assiduously, as evidenced by the differential uptake of SARS-CoV-2 testing. There's not much to be done about that, but the authors could consider controlling for some of those factors more clearly. As far as I can tell, only maternal age and parity were controlled for. Is there a different variable that might be a proxy for risk-taking behavior? If not, then perhaps this could be spelled out more in the discussion, particularly as it relates to how it could change interpretation of data. It's very easy to overestimate vaccine effectiveness when your active group (those that followed boosting guidance, without data on safety or benefit) could easily have done lots of other things during the Omicron wave to prevent infection (e.g. isolate completely).

The abstract confused me in the results section, between what was being compared. In one sentence (line 38), the third dose was compared to the second dose. In the next sentence (line 42), the third dose was compared to unvaccinated. This made it hard to tease out the main finding. This was a theme throughout. My recommendation is to focus on the additional benefit of the booster over just the primary series (2 doses), since this is really the question at issue as laid out in the introduction.

The use of the words 'substantial, significant, moderate, and severe' got jumbled up for me as a reader. I had to return many times to the definitions in Study Outcomes. I think the word 'significant' was where I got tripped up because it is used in other ways in the paper (e.g., for statistical comparisons). The authors might consider providing a reminder of what 'significant' means in the discussion. Also, they might consider removing the word "substantial" on line 235 because that threw me as well.

I also was confused by what "non-significant" hospitalization with COVID-19 might really mean, since

it was *not* pneumonia, hypoxia, or tachypnea. I'm guessing it was either coincidental positive test or observation for mild symptomatic COVID-19 where the patient was hospitalized anyway (perhaps because of the pregnancy, out of caution). The authors might consider discussing this nuance, since the reduction in 'hospitalization with SARS-CoV-2' could really just be a proxy for mild infection, in which case 43% protection isn't actually that bad and distracts from the bigger benefit against moderate/severe/critical illness.

Does it matter that people who deliver reach the end of follow up earlier than those that don't deliver? For example, people in their 3rd trimester might be more likely to be hospitalized (out of precaution) but also less likely to have been boosted (since they went on to deliver before they could get boosted). Perhaps the authors could control for trimester?

There's little discussion about how hybrid immunity or unknown prior infection might impact results. They say that people with prior infection are excluded, but that's only prior known infection and we know the unvaccinated group never gets tested. So, it would be good to comment more on how that could influence interpretation. (If the unvaccinated group is more immune than we realize, then perhaps it's no surprise that they look so similar to 2 shots during Omicron).

Minor points:

- Specify which guidelines (WHO, US, Israel?) recommend a third dose of COVID-19 for pregnant women (in Abstract and Discussion)
- The paragraph starting line 267 in the discussion was a little redundant with other parts of the discussion and could be shortened/cut.
- The point made starting line 275 about long-term effects of maternal infection are a little tangential. It's not addressed with this particular study. If included, I would link it to the fact that we don't know all the downstream consequences of infection and therefore any prevention of infection (even mild cases) during pregnancy may have unknown benefits.
- Figure 1, panel B. Consider adding a small legend or label specifying the color coding.
- Figure 2, panel A and B. Consider adding a label to each curve specifying the group.

Reviewer #2 (Remarks to the Author):

The authors reported the relative effectiveness of 3-dose vs. 2-dose of the Pfizer/BioN covid-19 vaccine vs. unvaccinated based on a population-based cohort analysis of pregnant women in Israel during the Delta and Omicron waves. Overall, the design and analyses of the study are clearly described and the paper is well written.

Major comments:

- 1) It is unclear whether individuals would continue to receive the 3rd dose if infection occurred after the 2nd dose. In another word, how much of the effect of the 3rd dose is actually a combined effect of natural immunity and vaccine immunity (aka hybrid immunity)? This might be a more pronounced issue during the Omicron period due to the high transmissibility of the variant and the higher percentage of asymptomatic infection.
- 2) It would be interesting to see whether the additional protection provided by the 3rd dose is associated with the length between the 2nd and 3rd dose.

We thank the reviewers for their careful reading of the text and constructive comments. We have amended the text accordingly. Below please find our point-by-point reply, embedded in the Reviewers Comments. In addition, we reviewed and polished the text (minor changes not marked).

Reviewer #1 (Remarks to the Author):

In this study, Dr. Guedalia and colleagues examined the potential benefit of a third dose of BNT162b2, an mRNA COVID-19 vaccine, in pregnancy. They note that such a third dose (or booster) is recommended for pregnant women, though there is not an established evidence base to support this recommendation, apart from extrapolating effectiveness findings from non-pregnant individuals.

As a randomized controlled trial cannot be done, the authors approach the problem by taking advantage of a large dataset available in Israel where vaccination history, pregnancy status, SARS-CoV-2 test results, and hospital outcomes are all linked. They also focused their analysis on two distinct periods defined by the predominant SARS-CoV-2 variant in circulation (e.g., Delta vs. Omicron BA.1) given the marked differences in the immune evasiveness of these variants.

The critical finding in this report is that a primary series (2 doses) was ineffective at preventing hospitalization with Omicron infection, and had much reduced effectiveness against hospitalization with moderate/severe illness compared to during the Delta wave. However, boosting with a 3rd mRNA dose rescued the protective effectiveness of vaccination, increasing protection against hospitalization with moderate/severe Omicron to 92%.

Strengths:

This is a critical study given (1) lack of data on boosting in pregnancy and (2) poor uptake of boosting across many populations. There is a dearth of understanding of how booster shots add significantly to a primary series for the prevention of bad outcomes in the setting of Omicron infection. While a primary series still provides a benefit on its own (particularly against severe disease and death), a primary series does not protect at all against Omicron infection (including mild and asymptomatic), for which the consequences in pregnancy may not be as benign. Robust data as presented in this report will be very helpful to underpin guidelines.

Weaknesses:

Like any effectiveness study, there is always the issue of the comparison groups not being quite the same people. Pregnant women who forgo the booster recommendation (and certainly those who don't get vaccinated at all) are different from those that follow recommendations assiduously, as evidenced by the

differential uptake of SARS-CoV-2 testing. There's not much to be done about that, but the authors could consider controlling for some of those factors more clearly.

As far as I can tell, only maternal age and parity were controlled for. Is there a different variable that might be a proxy for risk-taking behavior? If not, then perhaps this could be spelled out more in the discussion, particularly as it relates to how it could change interpretation of data. It's very easy to overestimate vaccine effectiveness when your active group (those that followed boosting guidance, without data on safety or benefit) could easily have done lots of other things during the Omicron wave to prevent infection (e.g. isolate completely).

A: We thank the reviewer for this question. This issue was discussed during study design, and partly the reason that we decided on using diagnosis of SARS-Cov2 during hospitalizations as an outcome, rather than reports of SARS-Cov2 positive lab tests. During the study period, all persons presenting to emergency departments or delivery rooms were screened for COVID-19 as a matter of protocol, regardless of the reason for admission. We designed the study to capture cases of illness requiring hospitalization as a better indicator of disease burden, and reduce selection bias as far as possible. Unfortunately, as a national database study, we do not have access to additional indicators of compliance to pandemic guidelines or other medical advice, or risk-taking or risk-averse behaviors. Please see Discussion section lines 250-259, where we described this approach, and additional reference to this issue that we included in the Limitation sections (lines 285-290).

The abstract confused me in the results section, between what was being compared. In one sentence (line 38), the third dose was compared to the second dose. In the next sentence (line 42), the third dose was compared to unvaccinated. This made it hard to tease out the main finding. This was a theme throughout. My recommendation is to focus on the additional benefit of the booster over just the primary series (2 doses), since this is really the question at issue as laid out in the introduction.

A: Thank you for pointing this out. We have revised the sentences mentioned to focus the presentation of results on the third dose vs. second dose comparison, as suggested (lines 40-44).

The use of the words 'substantial, significant, moderate, and severe' got jumbled up for me as a reader. I had to return many times to the definitions in Study Outcomes. I think the word 'significant' was where I got tripped up because it is used in other ways in the paper (e.g., for statistical comparisons). The authors might consider providing a reminder of what 'significant' means in the discussion. Also, they might consider removing the word "substantial" on line 235 because that threw me as well.

A: Agreed, this can be confusing. 'Substantial' at line 235 was deleted. We have added a sentence reprising the terms and their definitions to the Discussion section (lines 218-222). Additionally, we have minimized the use of 'significant'

throughout the study.

I also was confused by what “non-significant” hospitalization with COVID-19 might really mean, since it was *not* pneumonia, hypoxia, or tachypnea. I’m guessing it was either coincidental positive test or observation for mild symptomatic COVID-19 where the patient was hospitalized anyway (perhaps because of the pregnancy, out of caution). The authors might consider discussing this nuance, since the reduction in ‘hospitalization with SARS-CoV-2’ could really just be a proxy for mild infection, in which case 43% protection isn’t actually that bad and distracts from the bigger benefit against moderate/severe/critical illness.

A: Apologies for this misunderstanding – the intention was hospitalization for other indication, with incidental finding of positive COVID test on admission. We have added a phrase to clarify: hospitalized for any indication, with incidental diagnosis of SARS-CoV-2 infection to line 121-122 in the Study Outcomes section.

Does it matter that people who deliver reach the end of follow up earlier than those that don’t deliver? For example, people in their 3rd trimester might be more likely to be hospitalized (out of precaution) but also less likely to have been boosted (since they went on to deliver before they could get boosted). Perhaps the authors could control for trimester?

A: We thank the reviewer for the comment. The study group included only women who delivered during the study periods.

As for the potential bias that was stated by the reviewer, we further analyzed our data:

Third dose effectiveness calculated for the 3rd trimester by wave, against hospitalization with COVID-19 (third dose vs. second):

	3rd trimester (27 weeks-delivery)	Overall
Delta wave	93% (85-97)	92% (83-96)
Omicron wave	47% (36-56)	48% (37-57)

The vast majority of hospitalizations occurred during the third trimester (85.5% in Delta and 99.2% in Omicron), therefore we could not estimate vaccine effectiveness for the second trimester alone (insufficient number of cases). We have added a reference to this point in the revised manuscript (lines 206-207).

There’s little discussion about how hybrid immunity or unknown prior infection might impact results. They say that people with prior infection are excluded, but that’s only prior known infection and we know the unvaccinated group never gets tested. So, it would be good to comment more on how that could influence

interpretation. (If the unvaccinated group is more immune than we realize, then perhaps it's no surprise that they look so similar to 2 shots during Omicron).

A: This is a very cogent point, and indeed we considered this in our study design and analysis. Naturally, data on unreported or untested infections was unavailable. Individuals with asymptomatic or very mild infections were less likely to be tested, especially during the Omicron period, when there was no incentive to be tested, regardless of vaccination status or prior infection. This could also have contributed to the small difference observed during the Omicron wave. We have added a statement to the Limitations subheading (lines 285-291) that natural and hybrid immunity acquired in patients over the course of the pandemic, which may not have been captured by testing in the community, may have had differential protective effects in unvaccinated and vaccinated groups.

Minor points:

- Specify which guidelines (WHO, US, Israel?) recommend a third dose of COVID-19 for pregnant women (in Abstract and Discussion)

A: We have revised the Abstract and Discussion, adding citation of CDC and ACOG recommendations. Please see lines 27 (Abstract); 268-271 (Discussion).

- The paragraph starting line 267 in the discussion was a little redundant with other parts of the discussion and could be shortened/cut.

A: We have revised the indicated paragraph and the one before, to streamline this section (lines 268-275).

- The point made starting line 275 about long-term effects of maternal infection are a little tangential. It's not addressed with this particular study. If included, I would link it to the fact that we don't know all the downstream consequences of infection and therefore any prevention of infection (even mild cases) during pregnancy may have unknown benefits.

A: We have revised this paragraph as suggested, lines 276-281.

- Figure 1, panel B. Consider adding a small legend or label specifying the color coding.

- Figure 2, panel A and B. Consider adding a label to each curve specifying the group.

A: Suggested changes were made to the Figures.

Reviewer #2 (Remarks to the Author):

The authors reported the relative effectiveness of 3-dose vs. 2-dose of the Pfizer/BioN covid-19 vaccine vs. unvaccinated based on a population-based cohort analysis of pregnant women in Israel during the Delta and Omicron waves. Overall, the design and analyses of the study are clearly described and the paper is well written.

Major comments:

1) It is unclear whether individuals would continue to receive the 3rd dose if infection occurred after the 2nd dose. In another word, how much of the effect of the 3rd dose is actually a combined effect of natural immunity and vaccine immunity (aka hybrid immunity)? This might be a more pronounced issue during the Omicron period due to the high transmissibility of the variant and the higher percentage of asymptomatic infection.

A: Thanks for this important comment. Please see our response to Reviewer 1, above – indeed, hybrid immunity that may have been acquired and may have impacted our results. A sentence to this effect was added to the Limitations (line 285-290).

2) It would be interesting to see whether the additional protection provided by the 3rd dose is associated with the length between the 2nd and 3rd dose.

A: This is an interesting point, as the time elapsed between the 2nd and 3rd dose may associate with protection. We performed an additional analysis to answer this question. We calculated the risk for COVID-19 hospitalization during the Omicron period by intervals of 30 days and found that the time elapsed between vaccination had minimal to no effect on the risk for hospitalization. (There was insufficient incidence during the Delta period for meaningful analysis). During the Omicron period, when stratified by intervals between vaccination within the 30 days following eligibility, the risk for hospitalization was estimated as 1.36%, in the following 30 days – 1.46%, 1.55%, and 1.78%, for every sequential interval of 30 days, respectively. This was somewhat beyond the scope of the article; we leave it to the editors' discretion whether to add it to the manuscript.